# Search for Promising Strains of Probiotic Microbiota Isolated from Different Biotopes of Healthy Cats for Use in the Control of Surgical Infections

**DOI:** 10.3390/pathogens10060667

**Published:** 2021-05-28

**Authors:** Pavel Rudenko, Yuriy Vatnikov, Nadezhda Sachivkina, Andrei Rudenko, Evgeny Kulikov, Vladimir Lutsay, Elena Notina, Irina Bykova, Aleksander Petrov, Stanislav Drukovskiy, Ifarajimi Rapheal Olabode

**Affiliations:** 1Biological Testing Laboratory, Branch of Shemyakin-Ovchinnikov Institute of Bioorganic Chemistry of the Russian Academy of Sciences (BIBCh RAS), 142290 Pushchino, Russia; pavelrudenko76@yandex.ru; 2Department of Veterinary Medicine, Peoples’ Friendship University of Russia (RUDN University), 117198 Moscow, Russia; vatnikov@yandex.ru (Y.V.); petrov-ak@rudn.ru (A.P.); drukovskiy-sg@rudn.ru (S.D.); ifaradzhimi-ro@rudn.ru (I.R.O.); 3Department of Microbiology and Virology, Peoples’ Friendship University of Russia (RUDN University), 117198 Moscow, Russia; kulikov_ev@pfur.ru; 4Department of Veterinary Medicine, Moscow State University of Food Production, 125080 Moscow, Russia; vetrudek@yandex.ru (A.R.); recaro21@bk.ru (V.L.); 5Department of Foreign Languages, Peoples’ Friendship University of Russia (RUDN University), 117198 Moscow, Russia; notina-ea@rudn.ru (E.N.); bykova-ia@rudn.ru (I.B.)

**Keywords:** microbiota, biotope, probiotics, antibiotics, adhesion, antagonism, surgical infection, cats

## Abstract

Despite the introduction of modern methods of treatment, the creation of new generations of antibacterial agents, and the constant improvement of aseptic and antiseptic methods, the treatment of purulent–inflammatory processes remains one of the most complex and urgent problems in veterinary practice. The article presents the results of the isolation of indigenous microbiota from various biotopes of healthy cats, as well as the study of their biological marker properties for the selection of the most optimal strains in probiotic medicines for the control of surgical infections. It was demonstrated that isolated cultures of bifidobacteria and lactobacilli, which we isolated, revealed high sensitivity to antibiotics of the β-lactam group (excepting *L. acidophilus No. 24*, *L. plantarum “Victoria” No. 22*, *L. rhamnosus No. 5*, *L. rhamnosus No. 20,* and *L. rhamnosus No. 26*, which showed a significant variability in sensitivity to antibacterial drugs of this group, indicating the great potential of these microorganisms) and resistance to aminoglycosides, lincosamides, and fluoroquinolones (with the exception of gatifloxacin, which showed high efficiency in relation to all lactic acid microorganisms). The adhesive properties of the isolated lactobacteria and bifidobacteria were variable, even within the same species. It was found that the *B. adolescentis No. 23* strain of the *Bifidobacterium* genus, as well as the *L. plantarum No. 8*, *L. plantarum “Victoria” No. 22*, *L. rhamnosus No. 6*, *L. rhamnosus No. 26*, *L. acidophilus No. 12,* and *L. acidophilus No. 24* strains of the *Lactobacillus* genus had the highest adhesive activity. Thus, when conducting a detailed analysis of the biological marker properties of candidate cultures (determining their sensitivity to antimicrobial agents, studying the adhesive properties, and antagonistic activity in relation to causative agents of surgical infection in cats), it was found that the most promising are *L. plantarum “Victoria” No. 22*, *L. rhamnosus No. 26,* and *L. acidophilus No. 24*.

## 1. Introduction

The optimization of veterinary services, the reduction of morbidity and mortality of animals, and the effective prevention and improvement of measures to combat various diseases play a decisive role in improving the quality of life of animals [1,2,3,4,5,6,7,8]. At the same time, the population of small domestic animals, including cats, is growing every year; the number of shelters and nurseries for them is increasing, the number of veterinary clinics is growing, and new breeds of cats are bred and imported from abroad. In the practice of treating small domestic animals, including cats, purulent–inflammatory processes of soft tissues occupy one of the leading places [9,10,11]. At the same time, issues of the formation, progression, and prediction of the course of purulent–inflammatory processes in domestic animals, as well as their adequate treatment and prevention, remain poorly understood [12,13,14,15,16].

Treatment of pyoinflammatory processes to this day remains one of the most difficult and urgent problems in veterinary practice. Over the past decades, an increase in the number of pyoinflammatory diseases and infectious complications in animals has been recorded [17,18,19]. Despite the introduction of modern methods of treatment, the creation of new generations of antibacterial agents, the constant improvement of aseptic and antiseptic methods, and the number of cases of complications of surgical infection have not only not decreased, but, on the contrary, are increasing. This has led to the search for new, more effective methods of controlling purulent–inflammatory processes of soft tissues [10,20,21,22].

As a result of constant natural selection, an increasing number of complex and stable associations of microorganisms has emerged, which fill the various biotopes of the organism. Thus, in the course of evolution, microecological systems have emerged and populated the internal and external surfaces of the body, forming complex symbiotic complexes in them, which are the most stable and appropriate [23,24,25]. Microorganisms that live on the skin and mucous membranes are in a state of dynamic equilibrium with each other and the animal body. However, if unfavorable exogenous and endogenous factors exceed the compensatory capabilities of the synergistic system, “the host and his eubiosis”, then a change in the spectrum of microorganisms inhabiting it occurs, as a result of which microecological disorders occur, accompanied by immunodeficiency states, purulent–inflammatory complications, and other pathological processes in various organs and tissues [26,27,28,29].

In the process of evolution and as a result of the interaction between microbiocenoses and the host organism, the system of the host’s defense against various infections has improved. During the initial stages of evolution, indigenous microflora colonized the skin, mouth, respiratory, and digestive tracts of animals, thereby providing colonization resistance [30,31]. At the same time, other microbes, penetrating into the internal environment of the body, adapted to the new conditions of existence and acquired the characteristics of microbes–parasites. There are many facts regarding the phylogenetic relationship between microbiota, which in turn led to the emergence of more pathogenic microorganisms under the influence of unfavorable environmental conditions and multiple passages through susceptible animals [32,33]. Thus, in the process of evolution, various microorganisms have adapted to life and parasitism in animal tissues: viruses, bacteria, fungi, and protozoa. The mechanism by which the composition of the microbiota is controlled is a dynamic balance in the biotope maintained between obligate and facultative bacteria, as well as with the microflora of the environment; this is defined as “colonization resistance”, which is understood as a set of interrelated physiological, microbiological, and immunological factors that give stability to the indigenous microbiota and prevent the colonization of the animal’s body by foreign microorganisms [34,35].

One of the main functions of the indigenous microflora, which provide colonization resistance, is their adhesion to the receptors of the intestinal mucosa [36,37]. The ability of obligate autoflora to bind to mucosal receptors provides competition for them with pathogenic microorganisms [38,39,40]. Representatives of normal flora also have antagonistic activity against pathogenic and opportunistic microorganisms. The concept of antagonistic activity is very capacious: a high rate of reproduction, a wider range of enzymes, and the production of bactericidal and bacteriostatic substances [41,42,43,44,45]. Among the substances produced by lactobacteria and bifidobacteria, organic fatty acids occupy a special place, which have antagonistic activity against pathogenic microorganisms. They are capable of synthesizing bacteriocins (lactobrevin, lactocin, lactolin, lactocidin, plantaricin, colicin, helveticin, bulgaricin, and reuterin), natural antibiotic-like substances [46,47,48,49]. Additionally, representatives of normal microflora have natural resistance to antibacterial agents, which determines their use together with antibiotics [50,51,52].

In recent years, probiotics, bacterial preparations from living microbial cultures, have been widely used for the prevention and treatment of infectious diseases. Their use has led to an increase in the body’s resistance, favorable metabolic changes, and an antagonistic effect on microflora harmful to the animal. Probiotics do not cause side reactions, have no contraindications for use, and have a positive effect on the microbiocenoses of the biotopes of a macroorganism [10,17,20,46,53,54,55].

Therefore, the formation of the most optimal microbiocenoses of the skin and mucous membranes, their long-term preservation, and pre-emptive correction of emerging dysbacteriosis are the most important principles for maintaining the health of both individual animals and the species population in general. The purpose of this manuscript was to isolate the microbiota from various biotopes of healthy cats, as well as to study the biological properties for the selection of the most optimal strains in probiotic preparations for controlling surgical infection.

## 2. Results

A search was carried out in various biotopes of clinically healthy cats for promising strains of indigenous microorganisms: candidates for probiotic preparations for the treatment and prevention of surgical infection. The results of the conducted bacteriological analysis are shown in Table 1.

The data presented in the table indicate that in total, we isolated 214 cultures of microorganisms, most of which (102, 47.7%) were isolated from the contents of the intestinal tract of the studied animals. Most often, *E. coli* 63 (29.4%), *S. saprophyticus* 17 (7.9%), *S. faecalis* 16 (7.5%), and *L. acidophilus* 11 (5.1%) were isolated from different biotopes of cats.

In total, we isolated 37 (17.3%) strains of indigenous microorganisms, which were isolated only from a biopsy of the skin and samples of feces from the experimental animals. Thus, we most often isolated *L. rhamnosus* 6 (11.8%) and *L. acidophilus* 4 (7.8%) from the total number of isolated cultures from a skin biopsy. *L. plantarum* and *L. acidophilus* in seven (6.8%) cultures and *B. bifidum* and *L. rhamnosus* in four (3.9%) strains of the total number of isolated microorganisms were more often isolated from fecal samples. For a more thorough identification of the isolated *Escherichia coli* cultures, their serological typification was carried out, the results of which are shown in Table 2.

The data presented in the table show that the largest number of *E. coli* were isolated from fecal samples, namely, 37 (58.7%) microorganisms from the total number of isolated *E. coli* cultures. It should be noted that the isolated serogroups of *E. coli* isolates varied significantly among different biotopes of healthy animals. Thus, we most often isolated O111 (three, 30.0%) and O101 and O119 (two each, 20.0%) serogroups of *E. coli* from a skin biopsy; O116 (four, 28.6%), O113 (three, 21.4%), and O25, O114, and O119 (two each, 14.3%) serogroups from the oral cavity; O83 (eight, 21.7%), and O1, O4, and O22 (six each, 16.2%), and O2 and O18 (four each, 10.8%) serogroups from fecal samples; and O2 and O18 (one each, 50.0%) serogroups from peripheral blood samples of the total number of *Escherichia coli* cultures. It should be noted that none of the 63 *E. coli* isolates were classified as lactose-negative or hemolysin-producing *E. coli*. According to the results of the biological test, it was established that all microorganisms that were isolated from the contents of the oral cavity, feces, skin biopsy, and peripheral blood samples from 18 clinically healthy animals did not cause death of white mice.

Below, we present a detailed analysis of the biological marker properties of candidate cultures for the selection of the most promising strains in probiotic preparations for the treatment and prevention of surgical infections in cats.

First of all, the sensitivity of isolated strains of indigenous bacteria to antimicrobial drugs was determined. The sensitivity of isolated strains of microorganisms of the genus *Bifidobacterium* to antibacterial agents is shown in Figure 1 and Figure 2. Isolated cultures of *B. bifidum* showed the greatest sensitivity to drugs of the β-lactam group (penicillins and cephalosporins). Thus, the growth of four (100.0%) *B. bifidum* isolates was suppressed by benzylpenicillin concentrations in the range from 100 to 400; methicillin concentrations were suppressed in the range from 62.5 to 500; the concentration of amoxicillin was suppressed in the range from 12.5 to 25.0; the concentration of cefazolin was suppressed in the range from 12.5 to 50.0; the concentration of ceftriaxone was suppressed in the range from 6.25 to 25.0; cefepime concentrations were suppressed in the range from 1.56 to 3.12 μg/mL.

We noted a high degree of resistance of the isolated *B. bifidum* cultures in relation to the antibiotic of the aminoglycoside group, gentamicin. Thus, doses of gentamicin in the range from 1.56 to 25.0 μg/mL did not affect the experimental isolates of bifidobacteria; 100.0% of the strains were resistant to the antibiotic. However, a concentration of 50.0 μg/mL inhibited the growth of one (25.0%); a concentration of 100.0 μg/mL, two (50.0%); and a concentration of 200.0 μg/mL, three (75.0%) *B. bifidum* cultures. The MIC_50_ of gentamicin to *B. bifidum* isolates was 100.0, and MIC_90_ was 373.22 μg/mL.

We noted heterogeneous sensitivity in the isolated cultures of B. bifidum in relation to the antibiotic of the lincosamide group, lincomycin. Thus, doses in the range from 1.56 to 6.25 μg/mL did not affect the experimental strains; the concentration of lincomycin of 12.5 μg/mL inhibited the growth of one bifidobacteria (25.0%); a concentration of 50.0 μg/mL inhibited the growth of two bifidobacteria (50.0%); only the highest dilution of the antibiotic 100.0 μg/mL inhibited the growth of all four isolated cultures of bifidobacteria. The MIC_50_ of lincomycin against *B. bifidum* isolates was 60.99, and MIC_90_ was 786.22 μg/mL. It should be noted that the studied strains of *B. bifidum* were resistant to an antimicrobial agent from the group of fluorinated quinolones, enrofloxacin. Thus, the concentration of 6.25 μg/mL inhibited the growth of one bifidobacteria (25.0%); a concentration of enrofloxacin 12.5 and 25.0 μg/mL inhibited the growth of two bifidobacteria (50.0%); and three (75.0%) experimental cultures of bifidobacteria were sensitive to the lowest dilution of 50.0 μg/mL. However, none of the *B. bifidum* strains were resistant to gatifloxacin, a representative of a new generation of fluoroquinolones with anti-anaerobic properties (the MIC_50_ was 0.12, and the MIC_90_ was 0.47 μg/mL).

*B. adolescentis* strains isolated from clinically healthy cats also exhibited the highest sensitivity to antibiotics of the β-lactam group. Thus, the concentration of benzylpenicillin was from 200.0 to 400.0, amoxicillin from 12.5 to 25.0, methicillin from 125.0 to 500.0; cefazolin 12.5 to 50.0, ceftriaxone from 6.25 to 25.0, and cefepime from 1.56 to 3.12 μg/mL inhibited the growth of three (100.0%) experimental cultures of bifidobacteria. In addition, it was found that gatifloxacin was also a highly active agent for experimental isolates of *B. adolescentis*. The MIC_50_ of the preparation was 0.19, and the MIC_90_ was 1.57 μg/mL. It should be noted that *B. adolescentis* isolates exhibited heterogeneous sensitivity to gentamicin. Thus, all experimental strains were resistant at antibiotic concentrations in the range from 1.56 to 12.5 μg/mL; two (66.6%) isolates were stable at drug dilutions of 25.0 and 50.0 μg/mL; only at a minimal dilution of gentamicin of 200.0 μg/mL were all three isolated cultures of *B. adolescentis* susceptible to it. The MIC_50_ of gentamicin was 63.19, and the MIC_90_ was 492.38 μg/mL. *B. adolescentis* isolates are highly resistant to lincomycin. Thus, antibiotic concentrations from 1.56 to 6.25 μg/mL did not affect the experimental strains; at drug dilutions of 12.5 and 25.0 μg/mL, the sensitivity was recorded in one (33.3%) of the studied cultures; and the maximum concentrations of lincomycin of 50.0 and 100.0 μg/mL were active against two (66.6%) isolated cultures of bifidobacteria. The MIC_50_ of the antibiotic was 35.35, and the MIC_90_ was 459.68 μg/mL. We have also established a high resistance of experimental indigenous microorganisms to enrofloxacin. Thus, at maximum drug dilutions from 0.78 to 6.25 μg/mL, we recorded resistance in three (100.0%) studied strains; at a dilution of 12.5 μg/mL, two (66.6%) cultures of *B. adolescentis* possessed enrofloxacin resistance; at its maximum concentrations of 25.0 and 50.0 μg/mL, we found sensitivity in two (66.6%) cultures of ‘ifidobacterial. The MIC_50_ of enrofloxacin was 19.77, and the MIC_90_ was 154.27 μg/mL.

The antibacterial susceptibility of the isolated *L. acidophilus* cultures is shown in Figure 3.

Experienced strains of lactobacilli exhibit heterogeneous sensitivity to antibiotics of the β-lactam group. Thus, *L. acidophilus* isolates were susceptible to penicillin (MIC_50_ was 31.89, and MIC_90_ was 133.26 μg/mL), methicillin (MIC_50_ was 39.86, and MIC_90_ was 166.64 μg/mL), amoxicillin (MIC_50_ was 1.99, and MIC_90_ was 8.31 μg/mL), as well as to cefepime (the MIC_50_ of the drug was 0.26, and the MIC_90_ was 1.29 μg/mL). However, isolated strains of lactobacilli were resistant to antibiotics of the cephalosporin series I and III, generations of cefazolin and ceftriaxone, respectively. Thus, even at the maximum concentrations of cefazolin (25.0–50.0) and ceftriaxone (12.5–25.0 μg/mL), one (9.1%) strain of lactobacilli, namely, *L. acidophilus* No. 24, was resistant. The MIC_50_ of cefazolin was 12.60, and the MIC_90_ was 35.50 μg/mL. The MIC_50_ of ceftriaxone was 2.54, and the MIC_90_ was 18.86 μg/mL.

We also noted a high degree of resistance in the experimental cultures of lactobacilli to gentamicin, lincomycin, and enrofloxacin. Thus, 100.0% of the studied *L. acidophilus* strains were resistant at gentamicin concentrations ranging from 1.56 to 12.5 μg/mL; nine (81.8%) isolates were resistant at 25.0 μg/mL dilution of the drug; eight (72.7%) microorganisms showed antibiotic resistance at a dilution of 50.0 μg/mL. Even with a minimal dilution of gentamicin of 200.0 μg/mL, only six (54.5%) *L. acidophilus* isolates were sensitive to the drug. Lincomycin concentrations from 1.56 to 3.12 μg/mL did not affect the experimental strains; at drug dilutions of 6.25 μg/mL, the sensitivity was recorded in four (36.4%) isolated cultures of lactobacilli; at a dilution of 12.5 μg/mL, six (54.5%) isolates were sensitive; at dilutions of 25.0 and 50.0 μg/ml, we registered one (9.1%) strain of lactobacilli, namely, *L. acidophilus* No. 24, which was resistant to the drug; only at its maximum concentration of 100.0 μg/mL was the antibiotic effective against 100.0% of the studied bacteria. It was found that, at the maximum dilution of enrofloxacin in the range of 0.78–3.12 μg/mL, all 11 (100.0%) isolates showed resistance to the drug; even with the minimum dilutions of the antibiotic at 25.0 and 50.0 μg/mL, seven (63.6%) isolated cultures of *L. acidophilus* remained resistant. The MIC_50_ of gentamicin was 147.50, and the MIC_90_ was 1796.29 μg/mL. The MIC_50_ of lincomycin was 9.32, and the MIC_90_ was 36.67 μg/mL. The MIC_50_ of enrofloxacin was 78.50, and the MIC_90_ was 1100.19 μg/mL. It should be noted that *L. acidophilus* strains isolated by us showed sensitivity to gatifloxacin, a representative of a new generation of fluorinated quinolones. The MIC_50_ of gatifloxacin was 0.20, and the MIC_90_ was 0.46 μg/mL.

The sensitivity of isolated *L. plantarum* strains to antibacterial agents is reflected in Figure 4. Isolated *L. plantarum* strains showed heterogeneous sensitivity to penicillins. Thus, the experimental cultures were sensitive to methicillin, while antibiotic concentrations in the range from 125.0 to 500.0 μg/mL inhibited the growth of nine (100.0%) isolated bacteria. One (11.1%) strain of lactobacilli, namely, *L. plantarum “Victoria”* No. 22, remained resistant to benzylpenicillin at a concentration of 200.0 μg/mL; however, at the highest antibiotic dilution of 400.0 μg/mL, all nine (100.0%) experimental isolates were sensitive to it. It should be noted that resistance to amoxicillin, even at its maximum concentration of 25.0 μg/mL, was established in one (11.1%) isolate, namely, in *L. plantarum “Victoria”* No. 22.

We have also established a heterogeneous sensitivity of experienced cultured lactobacilli to cephalosporins. Thus, the concentration of cefazolin in the range from 0.78 to 3.12 μg/mL did not affect all the isolated strains; at a dilution of the drug of 6.25 μg/mL, resistance was found in seven (77.8%) cultures of lactobacteria; when using the antibiotic at 12.5 μg/ml, five (55.5%) isolates were susceptible to it; at a dilution of 25.0 μg/mL, sensitivity was found in six (66.7%) microorganisms; even at a concentration of 50.0 μg/mL, resistance to cefazolin was found in one (11.1%) isolated culture of *L. plantarum*, namely, *L. plantarum “Victoria” No. 22*. It should be noted that the L. plantarum “Victoria” No. 22 strain showed resistance to ceftriaxone and cefepime at concentrations of 12.5 and 1.56 μg/mL, respectively. However, the smallest dilutions of ceftriaxone 25.0 and cefepime 3.12 μg/mL inhibited the growth of all nine (100.0%) lactobacilli isolates. The MIC_50_ of cefazolin was 13.9, and the MIC_90_ was 55.84 μg/mL. The MIC_50_ of ceftriaxone was 5.46, and the MIC_90_ was 13.45 μg/mL. The MIC_50_ of cefepime was 0.68 and the MIC_90_ was 1.67 μg/mL.

Experimental cultures of *L. plantarum* show resistance to lincomycin. Thus, even at the maximum antibiotic concentrations of 50.0 and 100.0 μg/mL, we found resistance in one (11.1%) isolate, namely, *L. plantarum “Victoria” No. 22*. The MIC_50_ of lincomycin was 13.79 and the MIC_90_ was 67.54 μg/mL. We also established a high degree of resistance in the experimental isolates of lactobacilli to gentamicin and enrofloxacin. Thus, gentamicin concentrations in the range from 1.56 to 12.5 μg/mL did not retard the growth of nine (100.0%) isolated cultures of microorganisms; at a dilution of 25.0 μg/mL, we observed resistance in eight (88.9%) strains; at concentrations of 50.0 and 100.0 μg/mL, sensitivity was observed in four (44.4%) isolates; even at a minimal dilution of the antibiotic of 200.0 μg/mL, we found resistance in four (44.4%) isolated cultures of *L. plantarum*. The maximum enrofloxacin concentrations of 25.0 and 50.0 μg/mL inhibited the growth of five (55.5%) isolated strains of microorganisms. The MIC_50_ of gentamicin was 126.15, and the MIC_90_ was 1383.31 μg/mL. The MIC_50_ of enrofloxacin was 54.87, and the MIC_90_ was 1048.11 μg/mL. It is necessary to pay attention to the fact that we established a high sensitivity of *L. plantarum* isolates to gatifloxacin. Thus, the MIC_50_ of gatifloxacin was 0.13, and the MIC_90_ was 0.35 μg/mL.

All *L. rhamnosus* isolates were susceptible to benzylpenicillin (the MIC_50_ of the antibiotic was 55.3, and the MIC_90_ was 204.77 μg/mL), ceftriaxone (the MIC_50_ of the drug was 3.12, and the MIC_90_ was 8.98 μg/mL), and gatifloxacin (the MIC_50_ of fluoroquinolone was 0.24, and the MIC_90_ was 0.73 μg/mL). Isolated strains of lactobacilli showed heterogeneous sensitivity to cefepime. Thus, cefepime concentrations in the range from 0.04 to 0.18 μg/mL did not retard the growth of all 10 isolated bacterial cultures; at a dilution of 0.39 μg/mL, we observed sensitivity in three (30.0%) isolates; at a concentration of 0.78 μg/mL, the sensitivity was observed in seven (70.0%) isolated microorganisms; at a concentration of 1.56 μg/mL, antibiotic resistance was established in only one (10.0%) strain, namely, *L. rhamnosus No. 26*; however, all 10 (100.0%) tested lactobacilli were sensitive to the IV generation cephalosporin at a minimum dilution of 3.12 μg/mL. The MIC_50_ of cefepime was 0.56, and the MIC_90_ was 1.47 μg/mL. It was found that experimental *L. rhamnosus* isolates have resistance to cefazolin. Thus, even at a maximum concentration of the antibiotic of 50.0 μg/mL, we observed resistance to it in two (20.0%) isolated strains, namely, *L. rhamnosus No. 5* and *L. rhamnosus No. 26*. The MIC_50_ of cefazolin was 14.75, and the MIC_90_ was 72.47 μg/mL. In addition, we registered resistance in lactobacilli to amoxicillin. Thus, at the minimum dilutions of the antibiotic of 12.5 and 25.0 μg/mL, resistance was also observed in two (20.0%) experimental isolates, namely, *L. rhamnosus No. 20* and *L. rhamnosus No. 26*. The MIC_50_ of amoxicillin was 6.80, and the MIC_90_ was 30.26 μg/mL. A high degree of resistance was also registered in the experimental strains of microorganisms to methicillin, gentamicin, lincomycin, and enrofloxacin. Thus, even at the maximum methicillin concentrations of 250.0 and 500.0 μg/mL, three (30.0%) strains had resistance to it, namely, *L. rhamnosus No. 5*, *L. rhamnosus No. 20,* and *L. rhamnosus No. 26*. The MIC_50_ of methicillin was 139.47 and the MIC_90_ was 1455.76 μg/mL. At a minimum dilution of gentamicin of 200.0 μg/mL, only five (50.0%) studied cultures of lactobacilli were sensitive to the drug. The MIC_50_ of gentamicin was 179.89, and the MIC_90_ was 1372.70 μg/mL. At maximum lincomycin concentrations of 50.0 and 100.0 μg/mL, resistance was recorded in six (60.0%) isolated *L. rhamnosus* cultures. The MIC_50_ of lincomycin was 179.89, and the MIC_90_ was 1372.70 μg/mL. The maximum enrofloxacin concentration of 50.0 μg/mL also had no effect on six (60.0%) isolates of the experimental microorganisms. The MIC_50_ of enrofloxacin was 89.71, and the MIC_90_ was 1867.97 μg/mL (see Figure 5).

Thus, the cultures of bifidobacteria and lactobacilli isolated by us revealed a high sensitivity to antibiotics of the β-lactam group (with the exception of *L. acidophilus No. 24*, *L. plantarum “Victoria” No. 22*, *L. rhamnosus No. 5*, *L. rhamnosus No. 20,* and *L. rhamnosus No. 26*, which showed significant variability in sensitivity to antibacterial drugs of this group, which indicates the great potential of these microorganisms) and resistance to aminoglycosides, lincosamides, and fluoroquinolones (with the exception of gatifloxacin, which showed high efficacy against all lactic acid microorganisms).

Furthermore, the adhesive properties of the probiotic microbiota were determined. The results of determining the adhesive properties of strains of bacteria of the genus *Bifidobacterium* isolated from clinically healthy cats are presented in Table 3.

It can be seen from the data presented that the strains of bifidobacteria isolated by us had an average adhesive activity, since the AAI to erythrocytes of cats in the sample was 2.56 (from 1.56 to 4.24). It should be noted that the majority of the representatives of bifidoflora (four, 57.1%) isolates showed an average ability of adhesion, and only one (14.3%) isolated strain, namely, *B. adolescentis No. 23*, had highly adhesive properties, while the AAR index was 4.24 ± 0.41 tank./er.; K was 88.0%; and MAI was 4.81. The results of determining the adhesive properties of the isolated strains of bacteria of the *L. plantarum* species are shown in Table 4.

The presented data indicate that all *L. plantarum* isolates exhibit high and medium adhesion capacity, namely, five (55.6%) and four (44.4%) isolated strains, respectively. The AAI indicator, which in the sample is 4.40 (from 2.24 to 8.28), also indicates the highly adhesive properties of the isolated cultures of lactobacilli. It should be noted that *L. plantarum* No. 8 had the highest adhesive activity; the AAR index was 5.32 ± 0.26 bact/er., and for *L. plantarum* “Victoria” No. 22, the AAR index was 8.28 ± 0.17 bact/er., and the K index for these bacteria in both cases was 100.0%. It should be noted that we recorded high MAI values in these isolates, namely, for *L. plantarum* No. 8 it was 5.32, and for *L. plantarum “Victoria”* No. 22 it was 8.28. At the same time, in the tested cultures of *L. plantarum No. 8* and *L. plantarum “Victoria” No. 22*, we also observed the highest IR indices: 33 (24.8%) and 43 (20.3%) bacteria in the same field of view of the microscope, respectively.

The results of determining the adhesive properties of *L. rhamnosus* bacterial strains isolated from clinically healthy cats are reflected in Table 5.

The data in Table 5 indicate that only two (20.0%) *L. rhamnosus* cultures were classified as low-adhesive strains, while the rest of the microorganisms had medium and high adhesion capacity: four (40.0%) isolates each. The AAI index to erythrocytes of cats in the sample was 3.47 (from 1.36 to 6.16), which is regarded as the ability of adhesion in the isolated cultures of medium-grade microorganisms. The data given in the table also indicate that the *L. rhamnosus No. 6* strains had the greatest adhesive activity; at the same time, the SPA indicator was 5.04 ± 0.40 tank/er., the K indicator was 92.0%, and the MAI indicator was 5.47. For *L. rhamnosus No. 26*, the AAR index was 6.16 ± 0.51 baht/er., the K index was 88.0%, and the MAI index was 7.00. It should be noted that the IR index for these cultures was 19 (15.1%) and 36 (23.4%) bacteria in one microscope field of view, respectively.

The results of determining the adhesive properties of bacterial strains of the *L. acidophilus* species are shown in Table 6.

These data indicate that all *L. acidophilus* strains isolated from clinically healthy cats were capable of adhesion and were characterized by moderate to high adhesion. Thus, eight (72.7%) isolated cultures of lactoflora had highly adhesive properties, and the last three (27.3%) strains had an average ability to adhere. The AAI index in the sample of isolated cultures of microorganisms is 4.49 (from 3.72 to 6.24), which is regarded as a high adhesion capacity. We observed the highest rates of adhesive activity in *L. acidophilus strain No. 12*; the AAR index was 5.00 ± 0.33 bact/er., the K index was 96.0%, and the MAI index was 5.20. For *L. acidophilus* No. 24 isolate, the AAR indicator was 6.24 ± 0.46 bact/er., the K indicator was 92.0%, and the MAI indicator was 6.78. It should be noted that the IR index for these cultures was 29 (23.2%) and 33 (21.1%) bacteria in one microscope field of view, respectively.

The results of the study made it possible to reveal one of the main mechanisms of the protective action of the normal flora isolated by us, namely, their adhesive activity. It has been proven that the adhesive properties of the isolated lactobacteria and bifidobacteria are quite variable, even within the same species. It was shown that the highest adhesive activity was for the *Bifidobacterium* strain *B. adolescentis No. 23* genus, as well as the *Lactobacillus* strains *L. plantarum No. 8*, *L. plantarum “Victoria” No. 22*, *L. rhamnosus No. 6*, *L. rhamnosus No. 26*, *L. acidophilus No. 12,* and *L. acidophilus No. 24* genus.

The characteristics of the antagonistic activity of the isolated strains of indigenous bacteria to the main causative agents of surgical infection in cats are presented in Table 7. The data obtained indicate that antagonistic properties were inherent to varying degrees in all experimental strains of probiotic microorganisms, but the most active were representatives of the genera *B. adolescentis* (*n* = 3), *L. plantarum* (*n* = 9), and *L. acidophilus* (*n* = 11). Thus, 60 (23.5%), 197 (25.8%), and 291 (31.1%) cultures were sensitive to the isolates of these genera, and 170 (66.7%), 415 (54.2%), and 484 (51.8%) test cultures of microorganisms were highly sensitive, respectively, from the total number of test reactions performed. It should be noted that the lowest inhibitory activity was observed in representatives of the *B. bifidum* genus (*n* = 4) and *L. rhamnosus* genus (*n* = 10). Thus, 129 (37.9%) and 126 (14.8%) cultures were insensitive to the strains of these genera, and 105 (30.9%) and 142 (16.7%) test cultures, respectively, of the total number of test reactions performed were sensitive.

For greater clarity of the obtained material, as well as for a comparative analysis of the antagonistic activity of individual strains of indigenous bacteria within each species, we present the results obtained in separate tables. The results of determining the antagonistic activity of the isolated strains of bacteria of the *Bifidobacterium* genus to the main causative agents of surgical infection in cats are shown in Table 8.

It was found that the strains of the *B. adolescentis* species, namely, *B. adolescentis No. 1*, *B. adolescentis No. 23,* and *B. adolescentis No. 28*, were the most capable of eliminating both Gram-positive and Gram-negative microflora.

The results of determining the antagonistic activity of *L. plantarum* on the causative agents of purulent–inflammatory processes in cats are shown in Table 9.

The cultures of *L. plantarum No. 3*, *L. plantarum No. 8*, and *L. plantarum No. 27* also demonstrated relatively high growth retardation zones. Thus, *L. plantarum* isolate *No. 3* almost completely suppressed the growth of *S. intermedius* (growth delay zone—30.60 ± 1.07 mm), *S. pyogenes* (growth delay zone—32.00 ± 0.70 mm), and *K. pneumoniae* (growth delay zone—29.60 ± 0.50 mm); *L. plantarum* strain *No. 8* actively inhibited *S. pyogenes* cultures (growth delay zone—30.00 ± 0.31 mm) and *Escherichia coli* cultures (growth delay zone in the range of 18.40 ± 0.50 mm up to 26.20 ± 0.58 mm); *L. plantarum* isolate *No. 27* also actively eliminated both Gram-positive (growth delay zone in the range from 8.60 ± 0.50 to 20.80 ± 0.86 mm) and Gram-negative (growth delay zone in the range from 6.60 ± 0.50 to 30.40 ± 2.83 mm) microflora (Figure 6a,b). At the same time, the IIAA for these cultures was 20.8, 19.9, and 15.5 mm, respectively.

The least antagonistic activity was recorded in the probiotic *L. plantarum No. 25* strain on *S. uberis* (growth inhibition zone—2.40 ± 0.50 mm), *E. coli* O18 (growth inhibition zone—2.80 ± 0.37 mm), *P. aeruginosa* (growth inhibition zone—3.60 ± 0.50 mm), *E. aerogenes* (growth inhibition zone—1.00 ± 0.44 mm), *P. vulgaris* (growth inhibition zone—2.40 ± 0.50 mm), *P. mirabilis* (growth inhibition zone—2.00 ± 0.31 mm), and *C. albicans* (growth retardation zone—2.80 ± 0.37 mm) (see Figure 6b.). Only *S. intermedius, S. pyogenes*, and *E. coli O111* exhibited moderate sensitivity to it (the growth retardation zone in these cultures ranged from 11.60 ± 0.74 to 13.00 ± 0.54 mm). The IIAA index for *L. plantarum No. 25* was 6.2 mm.

The results of determining the antagonistic activity of isolated strains of *L. rhamnosus* bacteria on the main pathogens of purulent–inflammatory processes in cats are reflected in Table 10.

The research results given in the table indicate that the *L. rhamnosus* No. 26 strain was the most capable of eliminating both Gram-positive and Gram-negative microflora. Thus, this isolate inhibited the growth of coccal microflora in the range from 20.40 ± 0.50 to 30.00 ± 0.70 mm; *Escherichia coli* in the range from 23.40 ± 0.50 to 29.20 ± 0.86 mm (Figure 6c); *Pseudomonas aeruginosa*—31.40 ± 1.20 mm; *Klebsiella*—37.40 ± 1.32 mm; *Enterobacteriaceae*—27.00 ± 1.00 mm; citrobacters—15.40 ± 1.07 mm; proteus from 14.20 ± 1.49 to 16.80 ± 0.58 mm, as well as fungi of the *Candida* genus—22.20 ± 0.86 mm. In this case, the IIAA index was 24.6 mm. We also observed a relatively high antagonistic activity in the *L. rhamnosus No. 6*, *L. rhamnosus No. 29* and *L. rhamnosus No. 30* strains. Thus, the listed isolates inhibited the growth of Gram-positive microflora in the range from 16.40 ± 0.50 to 28.40 ± 0.92; Gram-negative bacteria, from 6.00 ± 0.70 to 32.60 ± 1.24 (see Figure 6c); and fungi of the *Candida* genus, from 16.00 ± 0.70 to 17.00 ± 0.70 mm. At the same time, the IIAA index for these strains was 22.4, 21.5, and 19.7 mm, respectively. It should be noted that L. rhamnosus No. 33 isolate shows the least antagonistic properties to *S. intermedius, S. epidermidis, S. faecalis, P. aeruginosa, K. pneumoniae, E. aerogenes, C. freundii, P. vulgaris, P. mirabilis,* and *C. albicans*; the growth retardation zones were 4.60 ± 0.51, 4.40 ± 0.50, 3.60 ± 0.50, 1.00 ± 0.54, 4.60 ± 0.50, 3.60 ± 0.50, 0, 0, 3.40 ± 0.50, and 1.80 ± 0.58 mm, respectively; and the AIAA index was 5.3 mm.

The results of determining the antagonistic activity of the isolated strains of bacteria of the *L. acidophilus* species against the main causative agents of surgical infections in cats are presented in Table 11.

These data indicate that among the bacteria of the *L. acidophilus* species isolated by us, the *L. acidophilus No. 24* strain had the greatest antagonistic activity. Thus, it actively inhibited the growth of Gram-positive (in the range from 20.20 ± 0.66 to 37.00 ± 1.64 mm) and Gram-negative (in the range from 17.40 ± 1.40 to 37.60 ± 0.92 mm) microflora, as well as representatives of *C. albicans* (18.20 ± 1.28 mm). At the same time, the IIAA indicator for it was 26.8 mm. We noted a relatively high intensity of antagonistic activity in the *L. acidophilus No. 12* isolate, which eliminated both Gram-positive (in the range from 19.60 ± 0.50 to 33.20 ± 2.03 mm) and Gram-negative (in the range from 12.20 ± 1.28 to 29.60 ± 1.72 mm) microflora, as well as fungi of the genus *Candida* (14.80 ± 0.86 mm), while the IIAA for it was 21.4 mm. It should be noted that *L. acidophilus No. 14, L. acidophilus No. 16,* and *L. acidophilus No. 21* had the least antagonistic properties. Thus, the claimed isolates moderately suppressed the growth of Gram-positive (in the range from 6.60 ± 0.50 to 12.60 ± 1.16 mm) and Gram-negative (in the range from 5.40 ± 0.92 to 13.80 ± 1.28 mm) bacteria, as well as representatives of *C. albicans* (in the range from 5.20 ± 0.58 to 5.80 ± 0.86 mm). In this case, the IIAA for them was equal to 9.8, 9.3, and 9.1 mm, respectively.

Thus, in order to select the most promising strains of probiotic microflora, we analyzed the indicators of the intensity of antagonistic activity, which were determined on the basis of comparing the values of the growth inhibition zones of pathogens of purulent–inflammatory processes in cats to candidate cultures. It was shown that among the tested probiotic strains, *L. plantarum “Victoria” No. 22, L. rhamnosus No. 26,* and *L. acidophilus No. 24* had the most pronounced antagonistic properties in which the IIAA indices were 26.0, 24.6, and 26.8 mm, respectively.

We also conducted studies to determine the correlation between the adhesive and antagonistic properties of strains of lactic acid microorganisms isolated from clinically healthy cats. It was found that the adhesive activity of isolated bifidobacteria and lactobacilli clearly correlates with their antagonistic properties (г = 0.9; *p* < 0.001 and г = 0.88; *p* < 0.001, respectively).

For a more detailed statistical analysis of the dependence of the level of the antagonistic activity of isolated lactic acid microorganisms on the level of their adhesive properties, we applied a regression analysis using a linear function, the results of which are shown in Figure 7.

The results of the performed regression analysis show that there is a clear correlation between the AAR level and the IIAA level. Thus, the regression equation for the indicators of AAR and IIAA in isolated bifidobacteria has the following form: y = 6.3129 * x −4.7469. It should be noted that the regression equation of the marker indices of AAR and IIAA in the isolated strains of lactobacilli also showed the presence of a rather significant pattern, which had the following form: y = 3.8282 * x +0.0294. It should be noted that the calculation of the reliability of certain mathematical models of the AAR levels in the representatives of bifidoflora and lactoflora isolated by us from the IIAA indicators showed the maximum level (*p* < 0.001).

In the future, to determine the severity of the antagonistic activity of industrial strains of lactic acid bacteria, as well as to select the concentration of lactobacilli in probiotic preparations, we developed a more accurate method for determining the level of quantitative antagonistic activity of lactic acid microorganisms, in which the method of serial dilution of the probiotic strain in MRS-2 is used, and sowing of test cultures is carried out on a two-layer solid nutrient medium. The results of determining the quantitative antagonistic activity of industrial strains of lactobacilli against the main causative agents of surgical infection in cats are shown in Table 12.

It can be seen from the above data that the studied *L. plantarum “Victoria” No. 22*, *L. rhamnosus No. 26,* and *L. acidophilus No. 24* strains had the highest quantitative antagonistic activity in relation to cultures of *S. pyogenes* and *P. aeruginosa*; namely, the MIC of these were 2.61 ± 0.04 log, 3.02 ± 0.08 log, 2.49 ± 0.04 log and 2.43 ± 0.03 log, 2.87 ± 0.07 log, 2.46 ± 0.03 log CFU/cm^3^, respectively. It should be noted that the studied strains of lactobacilli had high antimicrobial activity against all representatives of Gram-positive microflora. Thus, the MIC values in the experimental isolates of *L. plantarum* “*Victoria” No. 22, L. rhamnosus No. 26,* and *L. acidophilus No. 24* to test cultures of staphylococci and streptococci ranged from 2.49 ± 0.04 lg to 4.96 ± 0.06 lg CFU/cm^3^. However, it was found that industrial strains of lactobacilli have an unequal ability to influence the growth and viability of representatives of Gram-negative microflora. At the same time, MIK values varied over a wider range, namely, from 2.43 ± 0.03 log to 5.86 ± 0.09 log CFU/cm^3^. It was shown that the investigated *L. plantarum “Victoria” No. 22*, *L. rhamnosus No. 26,* and *L. acidophilus No. 24* strains have relatively low antifungal activity; the MIKA indices for *C. albicans* test cultures were 6.47 ± 0.10 lg, 6.74 ± 0.08 lg, and 6.44 ± 0.06 lg CFU/cm^3^, respectively. Thus, our proposed method for determining the level of antagonistic activity of lactic acid microorganisms makes it possible to carry out not only a qualitative but also a quantitative assessment of their antagonistic activity.

## 3. Discussion

Probiotic therapy has been widely used in the treatment and prevention of many infectious diseases and conditions in human and veterinary medicine for more than a decade [20,37,53,55]. Despite this, the problem of the development of purulent–inflammatory processes still remains one of the most complex and relevant in veterinary practice. This is due, as we believe, to the use of probiotics, which include strains of lactic acid microorganisms that are not particular to this type of animal.

Our preliminary studies have established that cats, with the development of surgical infection of varying severity isolate-specific pathogens, are not characteristic of other animal species [17,20]. In addition, representatives of the cat family have a natural resistance to a number of infections, which makes the probiotic microflora isolated from them attractive for research on its use in other animal and human species.

Based on the above, we conducted a search in various biotopes (skin biopsy, oral cavity contents, fecal samples, and peripheral blood samples) for promising strains of lactic acid microorganisms that are candidates for probiotic drugs in 18 clinically healthy cats.

Thus, in order to select the most promising strains for probiotic agents, we carried out a detailed analysis of the biological marker properties of candidate cultures, namely, the determination of their sensitivity to antimicrobial agents, the study of adhesive properties, and antagonistic activity in relation to the main causative agents of surgical infection in cats. On the basis of the conducted studies, the most promising strains for effective treatment and prevention of surgical infection in cats were selected, namely, *L. plantarum “Victoria” No. 22*, *L. rhamnosus No. 26,* and *L. acidophilus No. 24*. With the development of the latest technologies, a popular area of microbiological biotechnology is the creation of complex probiotics, which consist of several types of microorganisms. This direction is becoming promising because, complementing each other, these bacteria have a more effective therapeutic effect in comparison with monopreparations. Therefore, we carried out studies to assess the biocompatibility of the most promising strains, namely, *L. plantarum “Victoria” No. 22*, *L. rhamnosus No. 26,* and *L. acidophilus No. 24*. In co-cultivation, it was found that none of the three strains of lactobacilli are antagonists of each other. The data obtained reveal the possibility of assembling the selected strains of lactoflora in various combinations and thereby obtaining probiotic preparations with different properties.

## 4. Conclusions

The complex of microorganisms in the biotopes of an organism must be considered as an integral microbial ecosystem that plays an important role in the diagnosis of various pathological conditions. The development of unfavorable factors during purulent–inflammatory processes in animals entails a disruption of the microbial ecosystem and, accordingly, leads to a change in the balance between obligate and facultative microbiota. Probiotic microflora in the biotopes of the body forms its colonization resistance, which prevents the development of dysbiotic disorders that cause the occurrence of pathological conditions. Therefore, for a more effective fight against inflammatory processes, it is necessary to form optimally balanced microbiocenoses using probiotic biological products. The use of probiotic drugs in the complex treatment of purulent–inflammatory processes of soft tissues, in our opinion, is an evolutionarily justified approach, which requires further study in order to determine indications for widespread use in surgical veterinary practice.

The results of isolation of indigenous microbiota from various biotopes of healthy cats, as well as the study of its biological marker properties for the selection of the most optimal strains into probiotic preparations for combating surgical infection are presented. It was shown that the cultures of bifidobacteria and lactobacilli isolated by us revealed a high sensitivity to antibiotics of the β-lactam group (with the exception *of L. acidophilus No. 24, L. plantarum “Victoria” No. 22, L. rhamnosus No. 5, L. rhamnosus No. 20,* and *L rhamnosus No. 26*, which showed significant variability in sensitivity to antibacterial drugs of this group, which indicates the great potential of these microorganisms) and resistance to aminoglycosides, lincosamides, and fluoroquinolones (with the exception of gatifloxacin, which showed high efficacy against all lactic acid microorganisms). The adhesive properties of the isolated lactobacteria and ‘ifidobacterial were quite variable, even within the same species. It was found that the *B. adolescentis No. 23* strain of the *Bifidobacterium* genus, as well as the *L. plantarum No. 8*, *L. plantarum “Victoria”* No. 22, *L. rhamnosus No. 6*, *L. rhamnosus No. 26, L. acidophilus No. 12,* and *L. acidophilus No. 24* strains of the *Lactobacillus* genus have the greatest adhesive activity. Among the tested probiotic strains, *L. plantarum “Victoria” No. 22, L. rhamnosus No. 26,* and *L. acidophilus No. 24* had the most pronounced antagonistic properties, in which the IIAA indices were 26.0, 24.6, and 26.8 mm, respectively. It was found that the adhesive activity of isolated bifidobacteria and lactobacilli clearly correlates with their antagonistic properties (г = 0.9; *p* < 0.001 and г = 0.88; *p* < 0.001, respectively). Thus, when conducting a detailed analysis of the biological marker properties of candidate cultures (determining their sensitivity to antimicrobial agents, studying the adhesive properties, and antagonistic activity in relation to causative agents of surgical infection in cats), it was found that *L. plantarum “Victoria” No. 22*, *L. rhamnosus No. 26,* and *L. acidophilus No.24* were the most promising. In co-cultivation of *L. plantarum “Victoria” No. 22, L. rhamnosus No. 26,* and *L. acidophilus No. 24*, it was found that none of the three strains of lactobacilli were antagonists of each other. The data obtained reveal the possibility of assembling the selected strains of lactoflora in various combinations and thereby obtaining probiotic preparations with different properties.

## 5. Materials and Methods

### 5.1. Animal Subjects and Study Design

The cats were in a shelter for neglected animals in the Bogorodsky City District of the Moscow Region. The study included 18 clinically healthy mongrel, adult animals aged 2 to 5 years of both sexes. All of the animals were sterilized when they were admitted to the shelter. The cats were kept in separate cages so that the clinical condition, including appetite, could be monitored for each individual cat. The cats were fed a commercial dry, balanced adult animal feed, Purina Pro Plan Sterilized, three times a day. Prior to initiating the study, microscopic examination of feces after sugar centrifugation was performed on the feces of all cats to evaluate for parasite eggs, cysts, and oocysts. In the laboratory diagnosis of feces, macroscopic and microscopic methods and modern instrumental methods were used. Parasep (Apacor LTD, London, United Kingdom) disposable concentrators are designed for detecting the concentration of intestinal parasites by centrifugation through a specialized filter (a modification of the formalin ether method). Additionally, we used methods of flotation (floating), which are based on the difference in the specific weight of the flotation solution and the helminth eggs; the specific weight of the flotation solution is higher, and, as a result, the helminth eggs float to the surface in liquids and are found in the surface film. A commercially available direct fluorescent antibody assay was used to evaluate for the presence of *Giardia* spp. cysts and *Cryptosporidium* spp. oocysts.

In 18 clinically healthy cats, a search was carried out in various biotopes (skin biopsy, oral cavity contents, feces, and peripheral blood samples) for promising strains of lactic acid microorganisms: candidates for probiotic preparations. Promising strains were selected based on the characteristics of their biological marker properties: sensitivity to antibiotics, adhesive activity, and antagonistic properties.

To accumulate lactic acid microflora and increase the yield of bacterial mass, we used a solid nutrient medium “PSL” developed by us for the cultivation of lactobacilli, which contains glucose, blood, yeast autolysates (extracts), pancreatic hydrolyzate of casein, and microbiological agar.

Wealso carried out studies to determine the correlation between the adhesive and antagonistic properties of strains of lactic acid microorganisms isolated from clinically healthy cats; regression analysis was applied using a linear function.

### 5.2. Microbiological Research

When carrying out microbiological studies from the selected material isolated from cats with a Pasteur pipette, inoculations were made on nutrient media. For yeast-like fungi, Sabouraud’s glucose agar was used; for staphylococci, a peptone salt medium, yolk-salt agar, and MPA were used; for enterobacteria, Endo agar, Ploskirev’s medium, and bismuth sulfite agar were used; for bifidobacterial, Maurocyllmed medium and lactic acid were used. The inoculations were again incubated in a thermostat at 37–38 °C for 24 h, and in the absence of growth, the dishes were kept for up to 3 days.

After studying the cultural and morphological properties of all individual typical colonies, subcultures were made in the same test tubes and incubated at 37–38 °C for 24 h. The obtained pure cultures of bacteria were checked for mobility in preparations of a crushed drop using phase contrast microscopy in a darkened field of view and subjected to identification. For a quantitative bacteriological study of the contents of the oral cavity, fecal samples, skin biopsy, and purulent exudate or soft tissue biopsy were performed on 1.0 g of the studied substances in further studies. From the first test tube, which was considered a 10^−1^ dilution, further tenfold dilutions were prepared up to 10^−10^. Then, from each tube, 0.1 cm^3^ of the resulting mixture was inoculated into Petri dishes on the surface of solid nutrient media (Endo, MPA, Sabouraud, Ressel, Blaurocca nutrient medium, MRS, bismuth sulfite agar, yolk-salt agar, and solid nutrient medium “PSL”). Semi-liquid medium based on sodium thioglycolate (HiMedia, Maharashtra, India) creates anaerobic conditions. The crops were incubated in a thermostat at 37 °C for 7 days. The presence of microbial growth in nutrient media was assessed visually by the appearance of turbidity, film, sediment, and other changes. At the end of the incubation period, fixed preparations were prepared from test tubes with visible growth of microorganisms in color according to the Gram and Leffler method, followed by microscopy and identification.

The number of microorganisms in 1.0 cm^3^ of the starting material (C) was calculated by the following formula and expressed in logarithms with base 10:
C=(N/V)×K
where *N* is the average number of colonies in 1 bacteriological dish; *V* is the volume of the suspension, which is applied during inoculation on the surface of the agar; *K* is the multiplicity of dilution.

The morphology of bacteria was studied in smears stained according to Gram and Romanovsky–Giemsa staining. Further identification of biochemical properties was carried out in accordance with the “Bergey’s Identifier for Bacteria”. Gram-negative rods that gave a positive result in the test for the presence of catalase and a negative result in the test for cytochrome oxidase, oxidized and fermented glucose (in Hugh–Leifson’s medium), and reduced nitrates were assigned to the Enterobacteriaceae family. All isolated cultures were inoculated on Giss media with glucose, maltose, lactose, mannose, sucrose, mannitol, and dulcite. Gram-positive rod-shaped bacteria were additionally subcultured onto His medium with galactose, salicin, fructose, and arabinose. To determine the catalase activity of microorganisms, the bacterial mass removed with a loop from the agar surface was suspended in a drop of 3% hydrogen peroxide on a slide.

For further identification, the genus and species representatives of the Enterobacteriaceae family in the culture were subcultured onto Olkenitsky’s medium in a long variegated row, which included media with mannitol, maltose, sucrose, xylose, rhamnose, dulcite, sorbitol, salicin, Rochelle salt (d-tartrate), milk with litmus, and и beef-extract broth for the study of indole, as well as tests for the utilization of citrate, acetate, and the formation of H_2_S with methyl-mouth and the presence of phenylalanine deaminase. In Gram-negative rod-shaped bacteria, the fermentation of such carbohydrates as inositol and sorbitol was additionally determined using paper indicator systems (Nizhniy Novgorod, Russia); the utilization of sodium citrate and malonate; the production of hydrogen sulfide, indole, and acetylmethylcarbinol; the presence of ornithine decarboxylase, lysine decarboxylase, phenylalanine deaminase, and β-galactosidase enzymes. To eliminate motility in cultures of the *Proteus* genus, 96° alcohol was poured into bacteriological dishes with MPA before the studies, kept for 3–5 min, and then the alcohol was removed. Determination of *E. coli* serogroups was carried out using a set of “O-coliaglutinating serum” (“Armavir Biofabrika”, Russia).

To identify bacteria of the Pseudomonadaceae (Pseudomonas) family, the culture was subcultured on King B medium in a test tube beef-extract broth and grown in a thermostat at a temperature of 42 °C. For the differentiation of bacteria of the *Staphylococcus* genus from the genus *Streptococcus* genus, the presence of catalase was determined. The differentiation of the *Staphylococcus* genus from the *Micrococcus* genus used a glucose oxidation–fermentation test (Hugh–Leifson’s medium). To identify the species of bacteria of the *Staphylococcus* genus, tests were carried out for the presence of coagulase; the oxidation of mannitol, galactose, maltose, lactose, and sucrose; and the ability to grow in the presence of 10% NaCl. To identify the bacteria species of the *Streptococcus* genus, tests were carried out for the ability to grow in air at 10 °C and 45 °C at pH 9.6 in the presence of 6.5% NaCl, 40% bile; hemolysis; and sugar fermentation.

To determine the pathogenicity of isolated cultures, three white mice weighing 14–16 g were injected intraperitoneally with 1 billion mg. microbial cells for each strain of the microorganism. Laboratory animals were observed for 5 days. Cultures were considered pathogenic if one or more mice died within five days of infection. At the death of the animal, a bacteriological study of the pathogenic material selected from the parenchymal organs was performed to compare the isolate with the introduced pathogen.

### 5.3. Antibiotic Sensitivity and Statistical Analysis

Determination of the sensitivity of isolated microorganisms to antimicrobial agents was carried out by the method of serial dilutions in nutrient agar. A total of 10 antibiotics were used for test preparations (benzylpenicillin, methicillin, amoxicillin, cefazolin, ceftriaxone, cefepime, gentamicin, lincomycin, enrofloxacin, and gatifloxacin). The minimum inhibitory concentration (MIC) of the antimicrobial agent was determined by the delay in the growth of microorganisms when compared with the control in a cup containing the smallest amount of this drug. To compare the effectiveness of antibacterial agents, the calculation of MIC_50_ and MIC_90_ was performed using the probit analysis program (Probit100, Krasnoyarsk, Russian).

The program is designed for processing dose–effect curves, and allows one to calculate the effective dose for any level (LD1–LD99), as well as the corresponding confidence intervals for the 5% significance level.

### 5.4. Adhesive Properties

The determination of the adhesive properties of isolated strains of lactic acid microorganisms was carried out on a model of erythrocytes of clinically healthy cats. The study of adhesion was carried out using a light microscope (MBI-15-2); the adhesive properties were evaluated according to the following indicators: K—the coefficient of participation of erythrocytes in adhesion (% of erythrocytes that have adhesive bacteria on their surface); AAR—the average adhesion rate (the average number of microorganisms per 1 erythrocyte, which take part in adhesion); IC—index of contamination (the number of bacteria in one field of view of the microscope); IAM is the adhesiveness index of microorganisms, which was calculated by the formula: IAM = AAR × 100/K [56].

In addition, the AAI indicator was calculated—the average adhesion index (the arithmetic mean of the AAR in the sample). At least 25 erythrocytes were counted with no more than 5 erythrocytes in one field of view. The adhesiveness was considered to be zero at AAR from 0.0 to 1.0, low at AAR from 1.01 to 2.0, medium from 2.01 to 4.00, and high at more than 4.0.

### 5.5. Antagonistic Properties

The antagonistic activity of microorganisms was studied on solid nutrient media using the method of agar blocks [56]. To identify the severity of the antimicrobial activity of industrial strains of lactic acid bacteria and to select the concentration of lactobacilli in probiotic preparations, we developed a more accurate method for determining the level of quantitative antagonistic activity of lactic acid microorganisms, in which we used the method of serial dilution of the probiotic strain in MRS-2, and the inoculation of test cultures was carried out on a two-layer solid nutrient medium.

## Figures and Tables

**Figure 1 pathogens-10-00667-f001:**
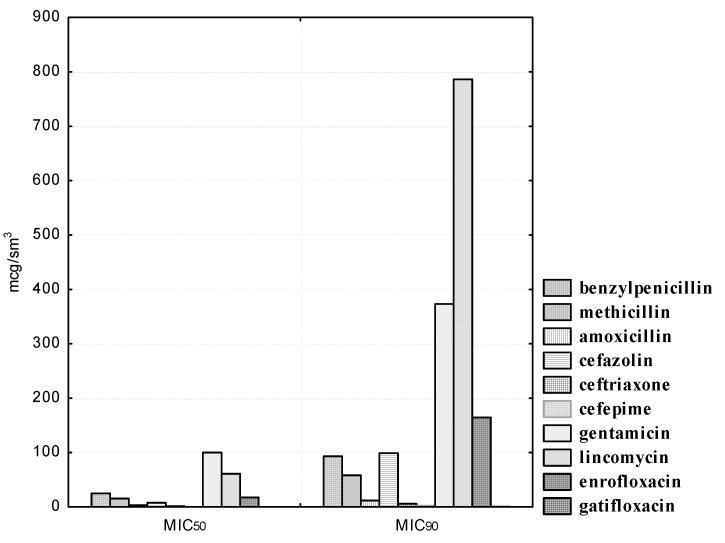
Antibiotic susceptibility of isolated *B. bifidum* cultures (*n* = 4).

**Figure 2 pathogens-10-00667-f002:**
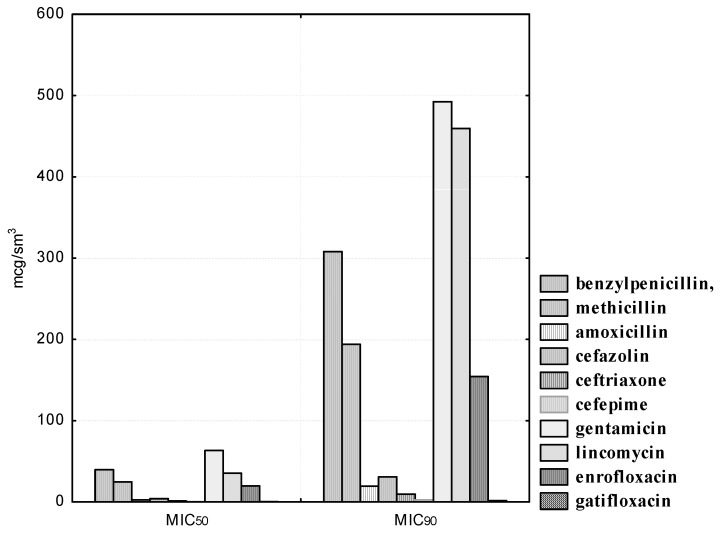
Antibiotic susceptibility of isolated *B. adolescentis* cultures (*n* = 3).

**Figure 3 pathogens-10-00667-f003:**
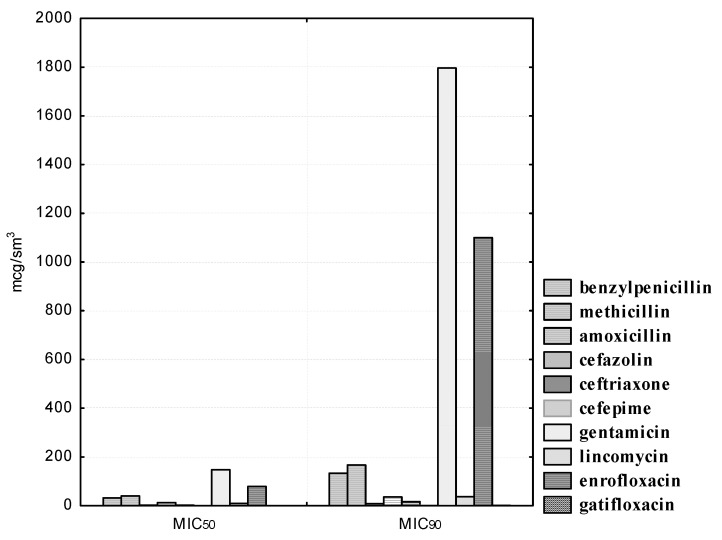
Antibiotic susceptibility of isolated *L. acidophilus* cultures (*n* = 11).

**Figure 4 pathogens-10-00667-f004:**
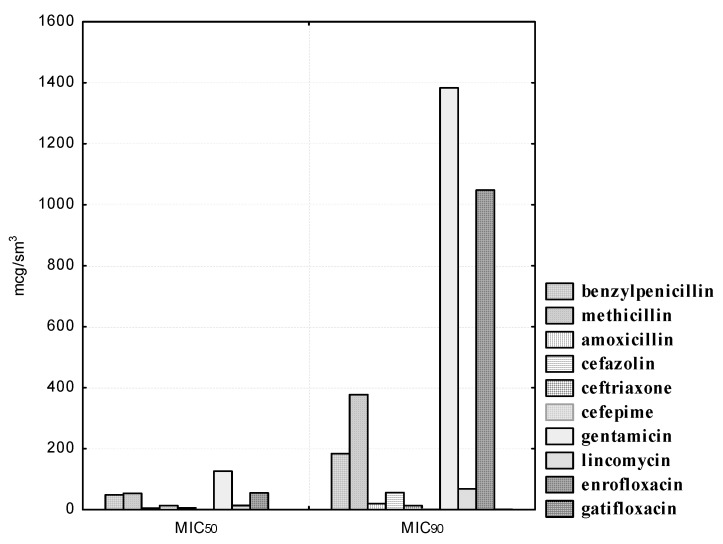
Antibiotic susceptibility of isolated *L. plantarum* cultures (*n* = 9).

**Figure 5 pathogens-10-00667-f005:**
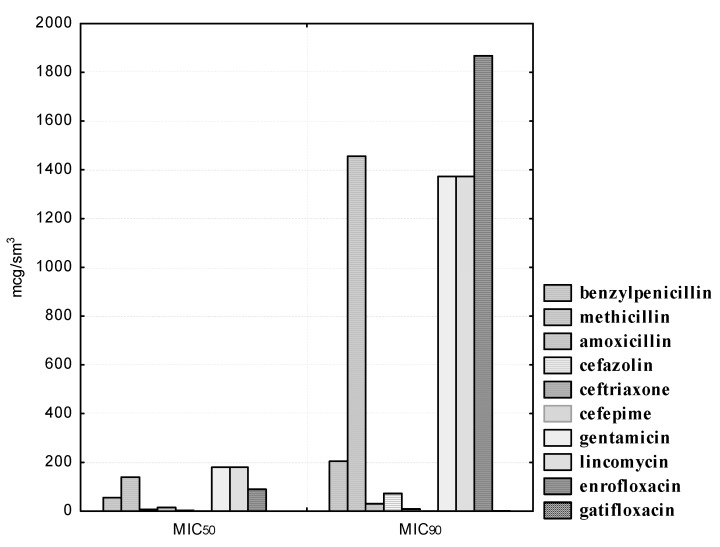
Antibiotic susceptibility of isolated *L. rhamnosus* cultures (*n* = 10).

**Figure 6 pathogens-10-00667-f006:**
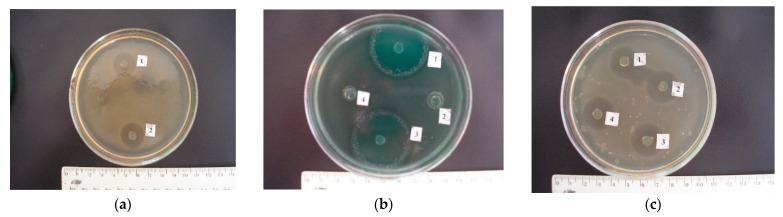
Antagonistic activity of Lactobacilli. (**a**): Antagonistic activity 1—L. plantarum No. 27; 2—L. rhamnosus No. 5 to S. aureus. (**b**): Antagonistic activity 1—L. plantarum No. 27; 2—L. rhamnosus No. 5; 3—L. rhamnosus No. 26; 4—L. plantarum No. 25 to P. aeruginosa. (**c**): Antagonistic activity 1—L. rhamnosus No. 6; 2—L. rhamnosus No. 26; 3—L. rhamnosus No. 29; 4—L. rhamnosus No. 30 to E. coli O18.

**Figure 7 pathogens-10-00667-f007:**
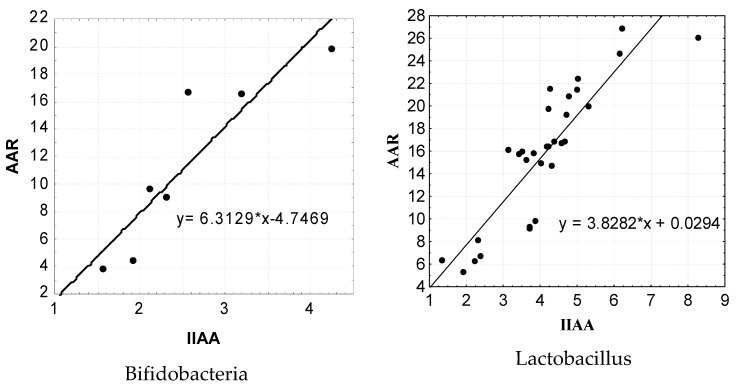
Results of regression analysis of the dependence of the adhesive properties of lactic acid bacteria on their antagonistic activity.

**Table 1 pathogens-10-00667-t001:** Results of bacteriological analysis of ecological niches of the organism of clinically healthy cats (*n* = 18).

Species of Microorganisms	Number of Isolates From:							
	Skin Biopsy		Contents of the Oral Cavity		Fecal Samples		Blood Samples	
	A.n.	%	A.n.	%	A.n.	%	A.n.	%
*E. coli*	10	19.6	14	23.7	37	36.4	2	100.0
*P. vulgaris*	3	5.9	–	–	–	–	–	–
*P. mirabilis*	2	3.9	4	6.8	–	–	–	–
*C. freundii*	–	–	–	–	8	7.8	–	–
*E. aerogenes*	–	–	–	–	9	8.8	–	–
*K. pneumoniae*	–	–	7	11.9	–	–	–	–
*P. multocida*	–	–	7	11.9	–	–	–	–
*S. aureus*	–	–	11	18.6	–	–	–	–
*S. saprophyticus*	12	23.6	–	–	5	5.0	–	–
*S. faecalis*	10	19.6	–	–	6	5.9	–	–
*S. agalactiae*	–	–	6	10.2	–	–	–	–
*S. pneumoniae*	–	–	10	16.9	–	–	–	–
*B. subtilis*	–	–	–	–	8	7.8	–	–
*B. adolescentis*	–	–	–	–	3	3.0	–	–
*B. bifidum*	–	–	–	–	4	3.9	–	–
*L. plantarum*	2	3.9	–	–	7	6.8	–	–
*L. rhamnosus*	6	11.8	–	–	4	3.9	–	–
*L. acidophilus*	4	7.8	–	–	7	6.8	–	–
*C. albicans*	2	3.9	–	–	4	3.9	–	–
Total	51	100.0	59	100.0	102	100.0	2	100.0

Note: indicates that the microorganism has not been isolated. A.n.-absolute number.

**Table 2 pathogens-10-00667-t002:** Serological typing of *E. coli* isolated from healthy cats (*n* = 18).

Serogroup	Number of Isolates From:							
	Skin Biopsy		Contents of the Oral Cavity		Fecal Samples		Blood Samples	
	A.n.	%	A.n.	%	A.n.	%	A.n.	%
O1	–	–	–	–	6	16.2	–	–
O2	1	10.0	–	–	4	10.8	1	50.0
O4	–	–	–	–	6	16.2	–	–
O9	–	–	–	–	3	8.1	–	–
O18	–	–	–	–	4	10.8	1	50.0
O22	–	–	–	–	6	16.2	–	–
O25	–	–	2	14.3	–	–	–	–
O26	–	–	1	7.1	–	–	–	–
O83	–	–	–	–	8	21.7	–	–
O101	2	20.0	–	–	–	–	–	–
O111	3	30.0	–	–	–	–	–	–
O113	–	–	3	21.4	–	–	–	–
O114	–	–	2	14.3	–	–	–	–
O116	–	–	4	28.6	–	–	–	–
O119	2	20.0	2	14.3	–	–	–	–
O127	1	10.0	–	–	–	–	–	–
O142	1	10.0	–	–	–	–	–	–
Total	10	100.0	14	100.0	37	100.0	2	100.0

Note: indicates that the microorganism has not been isolated.

**Table 3 pathogens-10-00667-t003:** Results of determining the adhesive properties of bacterial strains of the *Bifidobacterium* genus isolated from healthy cats (*n* = 18).

Isolated Strains	Indicators of the Adhesiveness of Microorganisms to Erythrocytes			
	C,%	AAR	IC, А.n. (%)	MAI
*B. adolescentis* No. 1	76.0	3.20 ± 0.44	27 (33.7)	4.21
*B. adolescentis* No. 23	88.0	4.24 ± 0.41	26 (24.5)	4.81
*B. adolescentis* No. 28	72.0	2.56 ± 0.37	16 (25.0)	3.55
*B. bifidum* No. 7	68.0	1.56 ± 0.27	14 (35.8)	2.29
*B. bifidum* No. 13	80.0	2.12 ± 0.31	11 (20.7)	2.65
*B. bifidum* No. 18	72.0	1.92 ± 0.29	13 (27.1)	2.66
*B. bifidum* No. 32	64.0	2.32 ± 0.39	18 (31.0)	3.62
AAI	2.56			

Note: hereinafter, C is the percentage of red blood cells that have bacteria on their surface; AAR—the average number of microorganisms per 1 erythrocyte, which are involved in adhesion; IC—index of contamination, the number of bacteria in one field of view of the microscope; MAI—microorganism adhesiveness index; AAI is the average adhesion index; A.n.-absolute number.

**Table 4 pathogens-10-00667-t004:** Results of determining the adhesive properties of *L. plantarum* bacterial strains isolated from healthy cats (*n* = 18).

Isolated Strains	Indicators of the Adhesiveness of Microorganisms to Erythrocytes			
	C,%	AAR, bac./er.	IC, abs.n. (%)	MAI
*L. plantarum* No. 3	88.0	4.80 ± 0.43	18 (15.0)	5.45
*L. plantarum* No. 8	100.0	5.32 ± 0.26	33 (24.8)	5.32
*L. plantarum* No. 10	84.0	3.44 ± 0.37	14 (16.3)	4.09
*L. plantarum “V”* No. 22	100.0	8.28 ± 0.17	43 (20.3)	8.28
*L. plantarum* No. 25	68.0	2.24 ± 0.36	17 (30.3)	3.29
*L. plantarum* No. 27	76.0	3.52 ± 0.46	24 (27.3)	4.63
*L. plantarum* No. 31	76.0	3.64 ± 0.46	28 (30.7)	4.78
*L. plantarum* No.34	92.0	4.32 ± 0.35	24 (22.2)	4.69
*L. plantarum* No. 37	84.0	4.04 ± 0.46	27 (26.7)	4.80
AAI	4.40			

**Table 5 pathogens-10-00667-t005:** Results of determining the adhesive properties of *L. rhamnosus* bacterial strains isolated from healthy cats (*n* = 18).

Isolated Strains	Indicators of the Adhesiveness of Microorganisms to Erythrocytes			
	C,%	AAR, bac./er.	IC, abs.n. (%)	MAI
*L. rhamnosus* No. 5	80.0	2.40 ± 0.31	14 (23.3)	3.00
*L. rhamnosus* No. 6	92.0	5.04 ± 0.40	19 (15.1)	5.47
*L. rhamnosus* No. 11	92.0	3.84 ± 0.38	18 (18.7)	4.17
*L. rhamnosus* No. 20	76.0	3.16 ± 0.42	24 (30.4)	4.15
*L. rhamnosus* No. 26	88.0	6.16 ± 0.51	36 (23.4)	7.00
*L. rhamnosus* No. 29	88.0	4.28 ± 0.41	24 (22.4)	4.86
*L. rhamnosus* No. 30	88.0	4.24 ± 0.40	29 (27.3)	4.81
*L. rhamnosus* No. 33	72.0	1.92 ± 0.31	11 (22.9)	2.66
*L. rhamnosus* No. 35	68.0	2.32 ± 0.37	19 (32.7)	3.41
*L. rhamnosus* No. 36	64.0	1.36 ± 0.26	7 (20.5)	2.12
AAI	3.47			

**Table 6 pathogens-10-00667-t006:** Results of determining the adhesive properties of *L. acidophilus* bacteria strains isolated from healthy cats (*n* = 18).

Isolated Strains	Indicators of the Adhesiveness of Microorganisms to Erythrocytes			
	C,%	AAR, bac./er.	IC, abs.n. (%)	MAI
*L. acidophilus* No. 2	80.0	4.60 ± 0.53	27 (23.5)	5.75
*L. acidophilus* No. 4	88.0	4.72 ± 0.45	20 (16.9)	5.36
*L. acidophilus* No. 9	88.0	4.20 ± 0.38	22 (20.9)	4.77
*L. acidophilus* No. 12	96.0	5.00 ± 0.33	29 (23.2)	5.20
*L. acidophilus* No. 14	88.0	3.88 ± 0.38	21 (21.6)	4.40
*L. acidophilus* No. 15	80.0	4.68 ± 0.53	30 (25.6)	5.85
*L. acidophilus* No. 16	88.0	3.72 ± 0.40	24 (25.8)	4.22
*L. acidophilus* No. 17	92.0	4.24 ± 0.38	20 (18.8)	4.60
*L. acidophilus* No. 19	88.0	4.40 ± 0.40	24 (21.8)	5.00
*L. acidophilus* No. 21	72.0	3.72 ± 0.53	25 (26.8)	5.16
*L. acidophilus* No. 24	92.0	6.24 ± 0.46	33 (21.1)	6.78
AAI	4.49			

**Table 7 pathogens-10-00667-t007:** Characteristics of the antagonistic activity of strains of probiotic bacteria to the causative agents of surgical infection in cats.

Bacteria	*n*	Test Cultures (*n* = 85), Which in Relation to Antagonists Were:							
		Not Sensitive		Insensitive		Sensitive		Highly Sensitive	
		A.n.	%	A.n.	%	A.n.	%	A.n.	%
*B. adolescentis*	3	9	3.5	16	6.3	60	23.5	170	66.7
*B. bifidum*	4	129	37.9	105	30.9	86	25.3	20	5.9
*L. plantarum*	9	35	4.6	118	15.4	197	25.8	415	54.2
*L. rhamnosus*	10	126	14.8	142	16.7	184	21.6	398	46.9
*L. acidophilus*	11	11	1.2	149	15.9	291	31.1	484	51.8

Note: not sensitive—growth inhibition zone 0–4 mm, insensitive—growth inhibition zone 5–9 mm, sensitive—growth inhibition zone 10–15 mm, highly sensitive—growth inhibition zone > 15 mm; A.n.-absolute number.

**Table 8 pathogens-10-00667-t008:** Antagonistic activity of bacterial strains of the *Bifidobacterium* genus to causative agents of surgical infection in cats.

Test Cultures	Strains						
	B. 1	B. 23	B. 28	B. 7	B. 13	B. 18	B. 32
*S. aureus* (*n* = 5)							
*S. intermedius* (*n* = 5)							
*S. epidermidis* (*n* = 5)							
*S. pyogenes* (*n* = 5)							
*S. uberis* (*n* = 5)							
*S. faecalis* (*n* = 5)							
*E. coli* O8 (*n* = 5)							
*E. coli* O18 (*n* = 5)							
*E. coli* O26 (*n* = 5)							
*E. coli* O111 (*n* = 5)							
*P. aeruginosa* (*n* = 5)							
*K. pneumoniae* (*n* = 5)							
*E. aerogenes* (*n* = 5)							
*C. freundii* (*n* = 5)							
*P. vulgaris* (*n* = 5)							
*P. mirabilis* (*n* = 5)							
*C. albicans* (*n* = 5)							
Note: B. 1—*B. adolescentis* No. 1, B. 23—*B. adolescentis* No. 23, B. 28—*B. adolescentis* No. 28, B. 7—*B. bifidum* No. 7, B. 13— *B. bifidum* No. 13, B. 18—*B. bifidum* No. 18, B. 32—*B. bifidum* No. 32.							
	—not sensitive (growth inhibition zone 0–4 mm),						
	—insensitive (growth inhibition zone 5–9 mm),						
	—sensitive (growth inhibition zone 10–15 mm),						
	—highly sensitive (growth inhibition zone > 15 mm).						

**Table 9 pathogens-10-00667-t009:** Antagonistic activity of bacterial strains of the *L. plantarum* on causative agents of surgical infection in cats.

Test Cultures	Strains								
	*L.* 3	*L.* 8	*L.* 10	*L.* 22	*L.* 25	*L.* 27	*L.* 31	*L.* 34	*L.* 37
*S. aureus* (*n* = 5)									
*S. intermedius* (*n* = 5)									
*S. epidermidis* (*n* = 5)									
*S. pyogenes* (*n* = 5)									
*S. uberis* (*n* = 5)									
*S. faecalis* (*n* = 5)									
*E. coli* O8 (*n* = 5)									
*E. coli* O18 (*n* = 5)									
*E. coli* O26 (*n* = 5)									
*E. coli* O111 (*n* = 5)									
*P. aeruginosa* (*n* = 5)									
*K. pneumoniae* (*n* = 5)									
*E. aerogenes* (*n* = 5)									
*C. freundii* (*n* = 5)									
*P. vulgaris* (*n* = 5)									
*P. mirabilis* (*n* = 5)									
*C. albicans* (*n* = 5)									
Note: L. 3—L. plantarum No. 3, L. 8—L. plantarum No. 8, L. 10—L. plantarum No. 10, L. 22—L. plantarum “Victoria” No. 22, L. 25—L. plantarum No. 25, L. 27—L. plantarum No. 27, L. 31—L. plantarum No. 31, L. 34—L. plantarum No. 34, L. 37—L. plantarum No. 37.									
	—not sensitive (growth inhibition zone 0–4 mm),								
	—insensitive (growth inhibition zone 5–9 mm),								
	—sensitive (growth inhibition zone 10–15 mm),								
	—highly sensitive (growth inhibition zone > 15 mm).								

**Table 10 pathogens-10-00667-t010:** Antagonistic activity of bacterial strains of the *L. rhamnosus* on causative agents of surgical infection in cats.

Test Cultures	Strains									
	L.5	L.6	L.11	L.20	L.26	L.29	L.30	L 33	L.35	L.36
*S. aureus* (*n* = 5)										
*S. intermedius* (*n* = 5)										
*S. epidermidis* (*n* = 5)										
*S. pyogenes* (*n* = 5)										
*S. uberis* (*n* = 5)										
*S. faecalis* (*n* = 5)										
*E. coli* O8 (*n* = 5)										
*E. coli* O18 (*n* = 5)										
*E. coli* O26 (*n* = 5)										
*E. coli* O111 (*n* = 5)										
*P. aeruginosa* (*n* = 5)										
*K. pneumoniae* (*n* = 5)										
*E. aerogenes* (*n* = 5)										
*C. freundii* (*n* = 5)										
*P. vulgaris* (*n* = 5)										
*P. mirabilis* (*n* = 5)										
*C. albicans* (*n* = 5)										
Note: L.5—L. rhamnosus No. 5, L.6—L. rhamnosus No. 6, L.11—L. rhamnosus No. 11, L.20—L. rhamnosus No. 20, L.26—L. rhamnosus No. 26, L.29—L. rhamnosus No. 29, L.30—L. rhamnosus No. 30, L 33—L. rhamnosus No. 33, L.35—L. rhamnosus No. 35, L.36—L. rhamnosus No. 36.										
	—not sensitive (growth inhibition zone 0–4 mm),									
	—insensitive (growth inhibition zone 5–9 mm),									
	—sensitive (growth inhibition zone 10–15 mm),									
	—highly sensitive (growth inhibition zone > 15 mm).									

**Table 11 pathogens-10-00667-t011:** Antagonistic activity of *L. acidophilus* bacterial strains against the causative agents of surgical infection in cats.

Test Cultures	Strains										
	L.2	L.4	L.9	L.12	L.14	L.15	L.16	L.17	L.19	L.21	L.24
*S. aureus* (*n* = 5)											
*S. intermedius* (*n* = 5)											
*S. epidermidis* (*n* = 5)											
*S. pyogenes* (*n* = 5)											
*S. uberis* (*n* = 5)											
*S. faecalis* (*n* = 5)											
*E. coli* O8 (*n* = 5)											
*E. coli* O18 (*n* = 5)											
*E. coli* O26 (*n* = 5)											
*E. coli* O111 (*n* = 5)											
*P. aeruginosa* (*n* = 5)											
*K. pneumoniae* (*n* = 5)											
*E. aerogenes* (*n* = 5)											
*C. freundii* (*n* = 5)											
*P. vulgaris* (*n* = 5)											
*P. mirabilis* (*n* = 5)											
*C. albicans* (*n* = 5)											
Note: L.2—L. acidophilus No. 2, L.4—L. acidophilus No. 4, L.9—L. acidophilus No. 9, L.12—L. acidophilus No. 12, L.14—L. acidophilus No. 14, L.15—L. acidophilus No. 15, L.16—L. acidophilus No. 16, L.17—L. acidophilus No. 17, L.19—L. acidophilus No. 19, L.21—L. acidophilus No. 21, L.24—L. acidophilus No. 24.											
	—insensitive (growth inhibition zone 5–9 mm),										
	—sensitive (growth inhibition zone 10–15 mm),										
	—highly sensitive (growth inhibition zone > 15 mm).										

**Table 12 pathogens-10-00667-t012:** Quantitative antagonistic activity of lactobacilli strains against causative agents of surgical infection, lg CFU/cm^3^.

Test Cultures	Probiotic Strains, lg CFU/cm^3^		
	*L. plantarum “Victoria”* No. 22	*L. rhamnosus* No. 26	*L. acidophilus* No. 24
*S. aureus* (*n* = 10)	3.27 ± 0.09	4.60 ± 0.07	4.18 ± 0.09
*S. intermedius* (*n* = 10)	3.58 ± 0.10	4.96 ± 0.06	4.33 ± 0.08
*S. epidermidis* (*n* = 10)	3.64 ± 0.06	4.39 ± 0.09	4.63 ± 0.06
*S. pyogenes* (*n* = 10)	2.61 ± 0.04	3.02 ± 0.08	2.49 ± 0.04
*S. uberis* (*n* = 10)	3.31 ± 0.07	3.58 ± 0.07	3.40 ± 0.06
*S. faecalis* (*n* = 8)	3.74 ± 0.07	4.04 ± 0.07	3.70 ± 0.06
*E. coli* O8 (*n* = 10)	3.91 ± 0.09	4.00 ± 0.07	3.76 ± 0.08
*E. coli* O18 (*n* = 5)	3.61 ± 0.11	3.91 ± 0.11	4.39 ± 0.07
*E. coli* O26 (*n* = 6)	3.47 ± 0.06	4.07 ± 0.10	4.12 ± 0.06
*E. coli* O111 (*n* = 10)	3.73 ± 0.07	4.06 ± 0.09	3.76 ± 0.08
*P. aeruginosa* (*n* = 10)	2.43 ± 0.03	2.87 ± 0.07	2.46 ± 0.03
*K. pneumoniae* (*n* = 10)	4.12 ± 0.07	3.85 ± 0.09	3.55 ± 0.09
*E. aerogenes* (*n* = 10)	3.70 ± 0.07	3.97 ± 0.07	4.00 ± 0.07
*C. freundii* (*n* = 10)	3.79 ± 0.08	3.97 ± 0.09	4.18 ± 0.06
*P. vulgaris* (*n* = 10)	5.26 ± 0.09	5.86 ± 0.09	5.71 ± 0.08
*P. mirabilis* (*n* = 8)	5.43 ± 0.09	5.73 ± 0.08	5.54 ± 0.07
*C. albicans* (*n* = 9)	6.47 ± 0.10	6.74 ± 0.08	6.44 ± 0.06

## Data Availability

Data is contained within the article.

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
