# Peer review of "Search for Promising Strains of Probiotic Microbiota Isolated from Different Biotopes of Healthy Cats for Use in the Control of Surgical Infections"

_pathogens, 2021, doi:10.3390/pathogens10060667_

Round 1
Reviewer 1 Report
-TITLE: indicate the purpose: surgical infection control
-Line 46-47: Are there articles that confirm what is indicated? Or is it professional casuistry?
-Line 50: Articles 9, 10, 12, 13 and 16 are not related to the feline family, so in this case they must speak of purulent-inflammatory processes in domestic animals.
Line 142: The number of animals in the study (it is done later), neither the breed, nor the average age are not indicated.
Wasn't a gross examination of the stool performed?. Indicate the technique used for the microscopic examination (flotation-centrifugation? = Modified Sheater). Indicate the commercial kits.
-Line 158: What product is yeast blood autolysate?.
-Line 164: What pathological material and where does it come from?
-Line 169: Thermostat is laboratory ovens????
-Line 171: The same tube media are used?
-Line 188.....:: Indicate after line 171.
-Line 216: 42ºC. how long?
-Line 223: What does this mean: at 10 ° С и 45 ° С. How long?
-Line 229: Where was the blood drawn from and what material was used to collect all the study samples?
-Line 239: Indicate on which the probit analysis is based and with what statistical program was it carried out?.
-Line 243: Indicate the method reference
-Line 258: N.S. Egorova: Reference?
-Line 327: Change MИC90 to MIC90
-Line 504: Cursive L. plantarum
-Table 7: The content of the note, indicate in material and methods.
-Results: In the text of the tables and figures, highlight the most significant and do not indicate everything that appears in the table / figure.
-The discussion does not exist and the conclusions is a summary of the results. Maybe unify the two points and especially review studies by other authors with whom to discuss.
Reviewer 2 Report
Reviewer’s comments and suggestions
In the current research paper, Rudenko et al. studied the various promising strains of probiotic microbiota that were isolated from different biotopes of healthy cats. The author reported that the most promising candidate cultures are L. plantarum «Victoria» â„–22, L. rhamnosus â„–26 and L. acidophilus â„–24. The paper content is good. However, the presentation was not enough to be accepted in this format of the manuscript. The manuscript showed consistency among the sections of the paper. The author can find my comments below and need to be incorporated in the revised version of the manuscript.
- Line 19-23, very big sentences, need to need to simplify the sentence
- Line 24-25 possible reason for this
- Line 34-35 putting bracket was not looking good
- Line 43 No need to put 1-8 references here, please check whether all references are suitable to cite here.
- Line 52-53 important lines need specific references
- Line 92-99 The sentences do not have references, pls specify.
- Line 132 Please discuss marker biological properties in the introduction. I have not seen the marker in the above.
- Line 141 Full affiliation needed
- All the figures need to be shown in terms of statistics. No error and significance was shown
- The discussion needs to elaborate based on the previous relevant studies cited in the paper. The results section n introduction seemed to be more exploratory.
- The author can minimize the section and only mention 4-5 lines about the novel result found in the study.
- All references need to be check according to the MDPI journal. I found all references have these issues.
Reviewer 3 Report
The manuscript titled “Search for promising strains of probiotic microbiota isolated from different biotopes of healthy cats” by Rudenko and coworkers and they have described the effects of selected strains in cats. I have few concerns regarding the present manuscript
-Isolation from cats is an interesting topic, however, the main sources of probiotics are others, why the authors focus on this specific issue?
In the same line, in my opinion, the introduction is too large and missing the main idea, please revise
-According to the authors, animal care is growing, but why there are searching strains in “health cats”, how the authors measure this
-Line 51, please check proinflammatory
-Lines 87, 92-93, 156, please modified for the most used word, microbiota
-Microbiological research needs more information about the culture methods, especially for anaerobic bacteria, in the same line, please summarizes this topic and add the manufacturer information
-Why the authors selected the erythrocytes model for adhesion, if they claim effects of these strains in microbiota, intestinal cells fits better in this topic
-Statistical analysis needs more detailed information
-The results section is too large, please summarizes the important information
-The discussion is missing
Round 2
Reviewer 3 Report
Dear authors, I read with interest your new manuscript and I grateful you take my comments as constructive and also, that that comment now appears in your new document, however, my principal concern is the adhesive test that you performed. You respond that ethical approval is necessary for that, but more and interesting data could be obtained with intestinal cells.
Author Response
Dear Reviewer, you are absolutely right. In our next experiment, we will definitely perform adhesion on the intestinal cells. Thanks a lot
Round 3
Reviewer 3 Report
Thank you to the authors for taking into account my previous comments